# Neurovascular Signaling at the Gliovascular Interface: From Flow Regulation to Cognitive Energy Coupling

**DOI:** 10.3390/ijms27010069

**Published:** 2025-12-21

**Authors:** Stefan Oprea, Cosmin Pantu, Daniel Costea, Adrian Vasile Dumitru, Catalina-Ioana Tataru, Nicolaie Dobrin, Mugurel Petrinel Radoi, Octavian Munteanu, Alexandru Breazu

**Affiliations:** 1Faculty of General Medicine, “Carol Davila” University of Medicine and Pharmacy, 050474 Bucharest, Romania; 2Department of Anatomy, “Carol Davila” University of Medicine and Pharmacy, 050474 Bucharest, Romania; 3Puls Med Association, 051885 Bucharest, Romania; 4Department of Neurosurgery, “Victor Babes” University of Medicine and Pharmacy, 300041 Timisoara, Romania; 5Department of Pathology, Faculty of Medicine, “Carol Davila” University of Medicine and Pharmacy, 030167 Bucharest, Romania; 6Clinical Department of Ophthalmology, “Carol Davila” University of Medicine and Pharmacy, 020021 Bucharest, Romania; 7Department of Ophthalmology, Clinical Hospital for Ophthalmological Emergencies, 010464 Bucharest, Romania; 8“Nicolae Oblu” Clinical Hospital, 700309 Iasi, Romania; 9Department of Neurosurgery, “Carol Davila” University of Medicine and Pharmacy, 050474 Bucharest, Romania

**Keywords:** adaptive coherence, brain energetics, gliovascular coupling, mitochondrial networks, thermodynamic criticality, redox plasticity, predictive coding, metabolic learning, entropy minimization, energy-information coupling

## Abstract

Thought processes in the brain occur as it continually modifies its use of energy. This review integrates research findings from molecular neurology, vascular physiology and non-equilibrium thermodynamics to create a comprehensive perspective on thinking as a coordinated energy process. Data shows that there is a relationship between the processing of information and metabolism throughout all scales, from the mitochondria’s electron transport chain to the rhythmic changes in the microvasculature. Through the cellular level of organization, mitochondrial networks, calcium (Ca^2+^) signals from astrocytes and the adaptive control of capillaries work together to maintain a state of balance between order and dissipation that maintains function while also maintaining the ability to be flexible. The longer-term regulatory mechanisms including redox plasticity, epigenetic programs and organelle remodeling may convert short-lived states of metabolism into long-lasting physiological “memory”. As well, data indicates that the cortical networks of the brain appear to be operating close to their critical regimes, which will allow them to respond to stimuli but prevent the brain from reaching an unstable energetic state. It is suggested that cognition occurs as the result of the brain’s ability to coordinate energy supply with neural activity over both time and space. Providing a perspective of the functional aspects of neurons as a continuous thermodynamic process creates a framework for making predictive statements that will guide future studies to measure coherence as a key link between energy flow, perception, memory and cognition.

## 1. Introduction

Neural activity creates rapid, distributed needs for ATP, O_2_, and substrate to allow for perception, attention, and memory. Hence, neural function is dependent on tightly coupled time-domain synchronization of the electrical (signaling) and biochemical (metabolic) fluxes at the gliovascular unit, an ultrastructural, nanoscale neuro-glio-vascular unit consisting of perivascular astrocytic endfeet, endothelial cells, pericytes, and the surrounding extracellular matrix (ECM) [1]. Biochemical signaling and mechanochemical transduction are utilized at this site to match local metabolic needs to vascular supply. Classical models of Neuro-Vascular Coupling (NVC), however, were primarily focused on feedforward reflex vasodilation generated by the activity-dependent release of soluble mediators into the synaptic cleft [2], and recent high-resolution imaging studies have shown that microvascular perfusion is affected by both activity-dependent responses and intrinsic vasomotor dynamics. Independent of synaptic blockade, low-frequency diameter oscillations in capillary and pre-capillary segments have been found to represent endogenous microvascular rhythmicity [3]. Additionally, astrocytic endfeet have been found to display both spontaneous and activity-dependent Ca^2+^ events that spread throughout the perivascular territory and modulate mural cell contraction, while pericytes have been found to exhibit Ca^2+^ oscillatory contractility that is regulated by Ca^2+^ released from IP_3_ receptors and ryanodine receptors and propagate to other segments via endothelial electrical conduction; further, neural activity can synchronize or phase-lock with such vasomotor rhythms, allowing for reciprocal coupling between neural computation and perfusion timing instead of a one-way relationship between neural activity and perfusion timing [4]. Astrocytes serve as the principal transduction pathways from synapses to vessels. Depolarization and neurotransmitter uptake at perisynaptic astrocytic processes activate Na^+^/K^+^-ATPases and Na^+^/Ca^2+^ exchangers generating localized Ca^2+^ microdomains that stimulate perivascular Ca^2+^ signaling. This Ca^2+^ signaling activates multiple enzymatic cascades producing vasoactive lipids from membrane phospholipids [5]. Moreover, K^+^ efflux through large-conductance channels results in hyperpolarization of pericytes and smooth muscle, adding an electrical component to chemical vasomodulation [6]. All of these processes occur within milliseconds post-synaptically and allow for interaction between ongoing oscillations and perfusion dynamics contributing to the preservation of rhythmic cortical states [7].

Metabolic communication between astrocytes and neurons occurs via energetic signaling. Glutamate uptake stimulates glycolytic metabolism and lactate production in astrocytes. Lactate produced in astrocytes is transported out through MCT1 and MCT4 and subsequently taken up by neurons through MCT2 and oxidized in mitochondria, thereby replenishing the necessary redox substrates required for synaptic vesicle cycling [8,9]. High-speed optical–metabolic studies show that lactate transients can occur before hemodynamic responses, indicating a feed-forward or predictive role in regulating supply [10].

Pericytes are perhaps the most obvious example of biochemical-to-mechanical conversion. Their expression of smooth muscle α-actin, myosin light chain kinase, and mechanosensitive ion channels enables pericytes to rapidly switch their contractile behavior [11]. Inter-pericytic gap junctions allow for synchronization of local Ca^2+^ spikes and modulation of microvascular resistance, permitting the propagation of dilation waves that can travel upstream to arterioles and provide localized pressure adjustments to optimize perfusion across active microdomains [12,13]. Pericyte and vascular oscillations are synchronous with infra-slow and delta cortical rhythms, suggesting that vascular mechanics play a role in organizing the temporal structure of global states such as sleep and sustained attention [14].

Shear stress-activated nitric-oxide synthase in endothelial cells produces nitric oxide that controls vascular tone and modulates glial and neuronal excitability in close proximity to the endothelium. Endothelial KATP channels couple redox state to depolarizing waves that regulate vasomotor control along the capillary tree [15]. Connexins 37, 40, and 43 facilitate retrograde transmission of these depolarizations over distances of hundreds of micrometers and coordinate synchronized dilation of capillaries on the order of seconds [16]. Such coordination allows capillaries to function as a coherent network regulating perfusion against varying demand. It is clear that there is evidence supporting continuous, graded regulation of CBF, regardless of the presence of rhythmic structure that can be correlated with neural activity; therefore, hemodynamic oscillations contain informative elements of neuro-glio-vascular control and do not simply reflect random variability [17,18]. More recently, advances in multi-omics and fast imaging techniques have allowed for insights into the regional specialization of astrocytic, pericytic, and endothelial programs [19]. Single-cell transcriptomics have revealed region-specific expression of transporters, aquaporins, enzymes, and mechanotransductive molecules in different regions of the brain, indicating the presence of region-specific energetic signatures: Astrocytes in sensory cortex express high levels of glycolytic enzymes and Ca^2+^ kinetic rates indicative of rapid metabolic response, while astrocytes in association cortex express high levels of oxidative enzymes and longer-duration signaling pathways; endothelial mechanotransductive complements may also vary depending on laminar flow regimes and could influence local vasomotor frequency structure [20].

Given that these regional specializations imply that metabolic signaling is tailored to the computational tempo of local circuits and is integrated with circuit function, the hierarchical relationship between metabolic and electrical rhythms is evident; infra-slow hemodynamic waves correlate with delta-band activity, while gamma-band activity correlates with finer vascular and astrocytic Ca^2+^ dynamics [21]. Phase coupling between cross-frequency bands provides a mechanistic link between global energetic states reflected in hemodynamic signals and local information processing reflected in faster oscillations [22]. Finally, studies using functional imaging and electrophysiology have shown that the temporal coherence between vascular and neuronal rhythms is a better predictor of behavioral performance than the power of neuronal activity alone, suggesting that cognition is dependent on maintaining supply–demand timing relationships in distributed networks [23]. Recent theoretical models formalize these observations by describing the gliovascular unit as a nonlinear control system that replicates the correlation between local field potentials and vessel diameter. If the delay between astrocytic Ca^2+^ signals and pericyte-mediated vascular responses exceeds critical threshold values, then regulation can transition from stable to unstable oscillation, similar to what is observed during fatigue or sleep deprivation; the critical zone is predicted to be most sensitive to demand changes with minimal energy expenditure, suggesting that cortical dynamics exist near equilibrium conditions where small timing disruptions in flow control result in large-scale network reorganization [24].

Recent methodologies are evolving toward greater temporal resolution. Sub-second correlations between astrocytic signaling and erythrocyte velocity have been detected using two-photon microscopy and genetically encoded reporters of oxygen tension and mitochondrial potential; functional ultrasound has mapped propagating flow waves across cortical columns; and fast volumetric optical systems have resolved 3D perivascular oscillations [25]. In conjunction with modeling, these studies suggest that cerebral flow patterns reside in low-dimensional attractor manifolds, consistent with the notion of organized dynamic landscapes underpinning cognitive states [26].

Thus, the microvasculature is not merely a passive delivery apparatus: it exhibits inherent rhythmic characteristics, integrates metabolic information, and significantly contributes to the preservation of the temporal relationship between neural demand and vascular supply—the “temporal architecture” of thought. Therefore, cognitive efficiency may rely more heavily on the accuracy of neuro-glio-vascular coupling than the absolute magnitude of energetics. Disruptions to the temporal alignment of supply and demand, and not necessarily frank energetic exhaustion, may disrupt pericyte function, astrocytic polarity, endothelial stiffness and desynchronize supply–demand timing relationships and thus make small disruptions in timing of flow potential early indicators of cognitive vulnerability [27,28].

Therefore, the aim of this study is to develop a conceptual model of the gliovascular interface as a timing regulator at multiple scales, integrating molecular, imaging, and computational evidence to provide a single, testable model of the gliovascular interface as a timing regulatory interface. By incorporating timing-based neuro-glio-vascular principles into electrophysiological models, this study seeks to elucidate mechanisms of how coupled glial and vascular oscillators contribute to resilient cognition, and how coherence may be restored once disrupted.

## 2. The Gliovascular Unit: Anatomy of a Signaling Ecosystem

Significant advances have been made in identifying the role of the gliovascular interface in homeostatic and regulatory mechanisms controlling CNS activity and the metabolic and hemodynamic regulation of neural computation. However, much is still unknown regarding the microvascular morphology as a highly dynamic signaling substrate for the integration of electrical, biochemical and mechanical stimuli [29,30]. Thus, the classical definition of the neurovascular unit (neurons—astrocytes—vessels) should be considered as a multi-cellular, polarized molecular network that is capable of integrating a wide range of signal types with high fidelity to both the spatial and temporal geometries of neural demand [31].

### 2.1. Microstructural Organization and Topological Logic

Perivascular astrocytic endfeet encase almost all cortical microvessels in a manner providing extensive ensheathment of cortical microvessels; however, coverage is heterogenous and often incomplete along capillary segments [32]. Astrocytic endfeet create a unique perivascular signaling layer, which enables intercellular communication by virtue of their internal ultrastructure, which contains discrete Ca^2+^ and lipid microdomains interconnected by endoplasmic-reticulum canals. Aquaporin-4, Kir4.1 and Connexin-43 are found in localized dense nanoclusters versus random distribution throughout; therefore, they serve as metabolic coupling sites for integrating ionic/osmotic signals and transforming biochemical gradients into coordinated hemodynamic responses [33,34].

Mechanical properties of the perivascular ECM beneath the endfoot sheath also exist. Specifically, the viscoelastic properties of the matrix and laminin isoform composition influence solute movement and the degree of mechanotransduction occurring at the endfoot-basal lamina interface [35]. Periodic release of metalloproteinases and glycosidases by astrocytes can cause temporary softening of the matrix facilitating rapid mechanical communication along perivascular pathways. This ability to soften may allow capillary walls to rapidly transmit pressure-linked and ionic perturbations and behave as resonant microstructures at low energetic cost [36]. Additionally, endothelial polarity exists and contributes to additional topological logic. Mechanosensing of shear stress (i.e., PECAM-1, Piezo1, GPR68) by luminal mechanoreceptors produces intracellular Ca^2+^ signals, whereas abluminal purinergic, prostaglandin and cannabinoid receptors are modulated by glial-derived signals [37,38]. Caveolae containing redox enzymes and NOS contain microdomains to detect oxygenation/metabolic states. Electrical continuity is provided to the entire vascular tree by endothelial gap junctions, which enable rapid depolarizing waves to be transmitted upward from arterioles toward the brain surface and provide retrograde coordination of flow across the cortex [39]. Pericytes positioned between endfeet and endothelium represent distributed effectors converting biochemical input into biomechanical capillary-tone regulation. Pericyte contractile apparatuses include alpha smooth muscle actin and non-muscle myosin II and are regulated by Ca^2+^-dependent and Rho kinase pathways [40]. Longitudinally interdigitated cooperative networks formed by pericyte processes are tonically regulated by astrocytic K^+^ flux, endothelial NO and mechanically induced signaling transduced via TRPV4 and pannexins [41].

### 2.2. Cellular Polarity and Molecular Asymmetry

Strong molecular asymmetry in gliovascular elements imparts directionality to signaling. Astrocytic polarity creates a relationship between activity near synapses and the perivascular compartment; Ca^2+^ signals produced near synapses propagate toward endfeet and connect to oscillatory NADH production and mitochondrial depolarization; these serve as intracellular metabolic messengers [42]. Cytoskeletal (actin and microtubules) structures promote vesicular trafficking of gliotransmitters/enzymes and aligned mitochondria along signaling paths reduce diffusion-limited delay and increase excitability–metabolism coupling efficiency [43].

The polarity of endothelial cells is complementary: luminal sensors transduce systemic variables (pressure, gas), while abluminal domains communicate signals to glia/neurons and allow bidirectional exchange between local neural state and systemic hemodynamics [44,45,46]. Systemic fluctuation can modulate endothelial signaling locally, and neural/glial activity can alter vascular resistance through chemoelectric waves traveling up the vascular tree creating a feedback loop connecting microvascular events to whole-brain hemodynamic regulation over very large dynamic ranges [47].

Similarly, pericytes exhibit polarity: proximal domains to the soma are enriched in mitochondria and sarcoplasmic reticulum and integrate Ca^2+^ oscillations, while distal process regions function as mechanotransductive sites converting wall tension into ionic flux [48]. This compartmentalization may enable phase-shifted flow oscillations that stabilize perfusion within local territories [28,49].

### 2.3. Communication Modalities and Integrative Signaling

Gliovascular element communication involves temporally ordered modalities. High-frequency ionic coupling (ms time-scale) includes K^+^ siphoning via Kir4.1, bicarbonate exchange, and diffusion of IP_3_/cAMP through gap junctions, allowing synchronization of activity among astrocytes and pericytes [50]. Slower chemical coupling uses gliotransmitters (ATP, adenosine, D-serine, glutamate); ATP activates P2Y receptors on endothelial/pericytic membranes causing relaxation of them via a Ca^2+^ response. In addition, mechanical coupling is another modality; deformation of the capillary due to capillary dilation/constriction changes the ECM and activates astrocytic mechanosensitive channels; in turn, pressure linked perturbations travel through perivascular space and computational models show that shear-deformation signaling travels many orders of magnitude faster than diffusion and thus allows coordination of timing of supply and demand over large distances at low metabolic cost [51,52,53]. Metabolic coupling integrates and interacts with the other modalities: lactate, pyruvate and CO_2_ transport through monocarboxylate/bicarbonate transporters provides a connection between glycolytic metabolism in astrocytes and oxidative phosphorylation in endothelial cells; synchronous NADH/FAD autofluorescent oscillations in astrocytes and pericytes indicate coupled redox cycling [54]. Organelle exchange between endfeet and endothelial cells via tunneling nanotubes has been shown to allow transfer of mitochondria and lysosomes during times of stress and enable cooperative regulation of energy supply and utilization [55].

### 2.4. Dynamic Remodeling and Adaptive Plasticity

Neural activity alters gliovascular structure and molecular composition. Activity increases perivascular endfoot coverage and enhances expression of aquaporin-4 and connexin-43; improved water handling and electrotonic coupling result from prolonged neural activation; conversely, reduction in sensory input or hypoperfusion causes endfoot retraction, decreases pericyte investment and reduces endothelial responsiveness [56,57,58]. These changes are reversible over days and occur indirectly through VEGF-, angiopoietin/Tie2- and ephrin-dependent cascades linking neural activity to vascular morphogenesis [59]. Experience-dependent vascular adaptation correlates with plasticity: angiogenesis typically occurs in concert with synaptogenesis; developing vessels follow developing dendrites/synapses; and astrocytic perivascular domains reorganize to stabilize newly forming synapses and potentiate [60,61]. It is possible that common molecular programs (including those regulated by BDNF and Wnt) will coordinate synaptic and vascular plasticity [49].

Chronic hypoxia/neuroinflammation may result in maladaptive remodeling leading to capillary rarefaction and impaired NVC [62,63]. Maintaining productive vs. destructive remodeling will be important for therapeutic interventions aimed at promoting cognitive resilience [64]. Table 1 lists the experimentally determined features of gliovascular microanatomy, polarity and modes of coupling, which illustrate its activity-dependent reorganization.

The gliovascular unit operates as an active networked control system for regulating electrical signaling, metabolic anticipation, and biomechanical vascular responses to maintain stable cerebral energetics. Rapid intercellular communication is supported by the microarchitecture of the gliovascular unit, especially across distances that diffusion cannot efficiently traverse [73,74]. Molecular polarity within gliovascular units provides feedforward and feedback directionality for neural and vascular compartmentalization; structural plasticity allows continuous recalibration of supply–demand timing under various behavioral and environmental conditions [75].

Pericyte tone changes, endothelial depolarizing waves, and astrocytic Ca^2+^ activity converge into coordinated vasomotor rhythms that correspond to energy delivery to information processing [76]. The nanoscale organization of channels, receptors, and cytoskeletal components enables local molecular events to produce coherent hemodynamic patterns, providing computation with a specialized, dynamically regulated circulatory substrate [77].

The purpose of this section is not to introduce hierarchical relationships but rather to emphasize the cooperative dependence of microstructure and signaling; microstructure regulates signaling, and signaling regulates microstructure and thus establishes a mechanistic relationship between capillary-level interactions and meso-scale flow oscillations and macro-scale stability necessary to support cognition.

## 3. Molecular Signaling at the Gliovascular Interface

The Glial–Vascular Interface is the site where glial cells use signaling mechanisms to control capillaries. The mechanism for this signaling occurs at the nanoscale level, with groups of molecules creating a “code” for converting synaptic information to capillary changes on a time scale of less than one second. We will be focusing on how this “logic” works, i.e., the “transduction logic,” which includes the cooperative action of receptor-channel complexes, lipid mediator systems, redox nanodomains and post-transcriptional regulatory programs to determine the timing and magnitude of flow responses [29,78].

### 3.1. Nanodomain Codes and Fast Transduction

Astrocytic perivascular membranes have dense nanodomains of metabotropic receptors, scaffolding proteins and ion channels located close enough together that they can interact with each other. These nanodomains include Gq/11-coupled mGluR5, P2Y1 and α1-adrenergic receptors all converging on the enzyme PLCβ. When activated, PLCβ cleaves PIP2 to generate two important signaling molecules: IP3 and DAG. IP3 binds to IP3R2 clusters on the surface of the ER to open them and allow Ca^2+^ to flow out into the cytoplasm. This creates a localized Ca^2+^ signal lasting anywhere from 10 milliseconds to several hundred milliseconds. Local phosphatases also help to sharpen the localized Ca^2+^ signal, as well as create a microenvironment between mitochondria and ER that helps to buffer the Ca^2+^ signal [79,80]. Store-operated entry (SOCE) through the channel Orai1 is maintained by the protein STIM1 to repeat the localized Ca^2+^ microtransient without causing a rise in Ca^2+^ in the entire cell, thus maintaining localization. TRPA1/TRPV4 channels provide mechanical Ca^2+^ entry that is dependent on the deformation of the ECM and to some degree independent of synaptic timing [81].

There are multiple levels of encoding in the nanodomains. PLA2 generates arachidonic acid from phospholipids that is then rapidly converted into EETs (P450 epoxygenases), prostanoids (COX) and 20-HETE in separate enzymatic domains that are regulated by the kinase state (PKA/PKC/ERK). Soluble epoxide hydrolase (sEH) regulates the rate of hydrolysis of EETs to dihydroxyeicosatrienoic acids (DHETs), which has been shown to match the temporal resolution of capillary relaxation to neuronal timing [82]. In addition, 2-AG (DAG-lipase) is generated in response to increased neural activity and activates the pericytic cannabinoid receptor CB1, resulting in a decrease in Ca^2+^ through Gi/o mediated signaling and results in a bias towards dilation [83]. Adenosine acts as a gain-control molecule: A2A receptors on the endothelium/mural cells cause dilation through cAMP-PKA signaling, whereas astrocytic A1 receptors cause tonic suppression of transmitter release and limit over-gain [84]. In addition, electrical/ionic coding is coordinated with lipid signals. Perivascular KIR-mediated K^+^ clearance creates localized hyperpolarization in the endothelium that is transmitted to the endothelium through connexins 37/40/43; the endothelial KIR2.1 channel (regulated by phosphoinositides) causes a non-linear wave of hyperpolarization to travel along the endothelial tree in response to small increases in K^+^ [85]. Rapid pH microdomains are generated by the Na^+^/HCO_3_^−^ cotransporter and carbonic anhydrase, and modulate the proton-sensitive conductance of pericytes on approximately 10 millisecond timescales, and thus affect the first hemodynamic phase prior to the slower equilibration of lipid molecules [86]. Redox control also plays a role in regulating the timing and amplitude of flow responses. Fluctuations in NADH/NAD^+^ and FAD in endfeet regulate nearby channels through redox-gated cysteines, modify substrate availability for NOS, and link transient ROS spikes to TRP opening and Ca^2+^ gain; catalase/peroxiredoxin kinetics terminate these events, and the geometry of cristae and tethering of ER to mitochondria confines multiple nanodomains with different thresholds/filters within one endfoot [87,88].

### 3.2. Effector Integration: Mural Mechanics, Endothelial Relays, and Blood-Borne Signals

Pericytes generate biomechanical outputs from their nanodomain inputs through coupled programs involving RhoA-ROCK and MLCK. Ca^2+^—calmodulin—MLCK drives the rapid contraction of pericytes, while ROCK stabilizes tone by inhibiting myosin phosphatase (MYPT1 phosphorylation), such that brief Ca^2+^ spikes favor plastic dilation, whereas sustained RhoA leads to slowly relaxing constriction [89]. Mechanical force onto the basement membrane is determined by focal adhesions (integrin–talin–vinculin); PKA/PKG phosphorylation reduces adhesion and strengthens dilation when Ca^2+^ thresholds are met, establishing a state-dependent “adhesion memory” [90]. The endothelium scales local control into network coherence. Activation of Piezo1 by shear forces generates Ca^2+^ signals that increase the phosphorylation of eNOS (Ser1177 via Akt/AMPK), raising NO within seconds; NO diffuses to mural cells and increases KATP-dependent hyperpolarization spread [91,92]. The degree of sulfation of the glycocalyx (syndecan/glypican) determines the threshold for mechanotransduction and links erythrocyte flux to depolarization; disassembly of caveolin-1 limits eNOS-calmodulin access and, during high-frequency demand, temporarily increases NO output [93].

EETs derived from P450 in endothelial cells act as EDHFs, sustaining dilation when NO is limited due to diameter expansion [94]. Blood signals include the following: deoxygenated RBCs release ATP (pannexin-1) to endothelial P2Y receptors, increasing Ca^2+^/NO; hemoglobin scavenges NO but nitrite reduction to NO at low PO_2_ prolongs dilation in hypoxic areas; platelet sphingosine-1-phosphate via S1PR1 maintains barrier integrity and eNOS coupling, linking coagulability to NVC efficiency [94]. Vasomotor tone is also influenced by perivascular nerves: CGRP/VIP causes dilation via cAMP-PKA, while NPY via Y1 causes constriction; norepinephrine acting on astrocytic α1 receptors increases the frequency of Ca^2+^ microdomain spikes and establishes the basal vasomotor tempo during wakefulness [95,96,97]. Gasotransmitters also work cooperatively: NO (endothelium/interneurons), CO (astrocytic heme-oxygenase), and H_2_S (CBS) activate sGC and KATP; low H_2_S increases sGC sensitivity to NO, while CO extends the cGMP half-life via PDE modulation, thus extending the effective window of each dilatory event [98]. Modulatory slow inputs maintain the output through state transitions [99]. Slow, long-term tuning readjusts the thresholds for future demands. Astrocytic AMPK detects AMP/ATP shifts, increases glycolytic production and increases the translocation of perivascular MCT to decrease latency between synaptic activity and substrate export; SIRT3/SIRT1 broaden the antioxidant/redox dynamic range and modulate the lipid-enzymatic coupling without saturating Ca^2+^ pathways [100]. Mild hypoxia activates HIF programs to increase endothelial VEGF and mural PDGFR signaling to regulate capillary density and pericyte coverage over days; post-transcriptional nanocontrol is provided by miR-132, miR-29 and miR-143/145 regulating tight junctions, contractile proteins and lipid enzymes, with miR-132 downregulating eNOS inhibitors while preserving barrier turnover [101,102]. Astrocytic lncRNA regulates the composition of nanodomain lipid/Ca^2+^ constituents toward dilation or constriction depending upon the recent history of activity, and phosphorylated eIF2α transiently suppresses endfoot translation during stress to maximize signaling energy, and restore synthesis during quiescence [47,103,104,105]. Figure 1 illustrates how the gliovascular signaling streams to and from the rapid ionic event-centric, to those modulations that are much slower and adaptive, preserving long-term coherence.

Daily circadian rhythms also modulate endothelial redox rhythms, astrocytic glycogen metabolism, lipid-enzyme/connexin expression, and gap-junction coupling via REV-ERB/ROR pathways, resulting in daily variations in latency/magnitude and coupling strength [106,107,108]. Finally, there is surface-level regulation adding minutes-to-hours of tuning: Cx43 phosphorylation/trafficking alters the number/lifetime of plaques; IP3R phosphorylation and S-nitrosylation establish the probability of release of Ca^2+^ from nanodomains; glycocalyx thickness/sulfation alters shear-induced eNOS sensitivity; and ROCK/phosphatase balance in murals adjusts the capillary viscoelastic transfer functions [109,110,111,112].

Collectively, these layers represent a predictive controller: fast localized ionic/Ca^2+^ events align early dilation, lipid/gasotransmitter cascades extend it, and metabolic/gene programs readjust the thresholds so future demands fall within an optimal dynamic range [113]. Disruptions to any layer (loss of redox containment, glycocalyx degeneration, focal adhesion disruption) would lead to timing errors, rather than solely amplitude loss, which is consistent with cognitive vulnerabilities in vascular/inflammatory diseases [114].

Therefore, a timing-centered molecular model represents a useful complement to anatomical/system-level models, and may explain why slight molecular alterations can produce cognitive dysfunction and why restoring synchrony improves function even after structural damage [115].

The major timing-based claims are supported by a wide range of empirical evidence across multiple levels of scale. The high-speed in vivo imaging studies have demonstrated that there is rapid coordination (subsecond) between perivascular astrocytic Ca^2+^ microdomains, mural cell contraction, and changes in both capillary diameter and red blood cell velocity, which suggest that behaviorally relevant timeframes exist for regulating microvascular control and thus microvascular regulation may be more than just delayed metabolic aftereffects [116]. Pericyte and capillary segment intrinsic low-frequency vasomotor oscillations can occur independently of synaptic blockade and are able to induce entrainment of neighboring vessels through endothelial electrical conduction suggesting the existence of an endogenous microvascular rhythmicity capable of being phase-locked to neural activity [117]. At the network level, infralow hemodynamic waves have correlated with delta band potentials, and faster vascular/Ca^2+^ dynamics have correlated with gamma band states; cross-frequency coupling between these energy-related and electrophysiology-related rhythms has been observed using optical, electrophysiological, and functional imaging methods [118]. While none of this data provides a singular mechanism for controlling the nervous system, it does provide evidence supporting the idea that gliovascular timing is not random or unstructured and that it is dependent upon the current state of the nervous system as well as reciprocal interactions between the nervous system and gliovascular function.

Critique and Alternative Perspectives

A major point of debate is whether rapid NVC is primarily initiated by astrocyte-nanodomain-driven signaling or by direct neuronal/endothelial signaling. There are proponents who suggest that astrocytic Ca^2+^-lipid nanodomains are the primary initiators of tone change, while others emphasize rapid neuronal NO signaling and endothelial electrical conduction, suggesting that astrocytes play a greater role in slower gain control and metabolic support [119]. While evidence suggests that both pathways contribute to rapid NVC, the relative contribution of each pathway is dependent on vessel type, location, and state. Our framework unites these pathways into a timing-based hierarchical structure and provides testable predictions regarding the contextual dependence of control.

## 4. Neurometabolic Integration and Electrophysiological Coupling

In addition to coordinating their ionic and metabolic activities, the elements of the neuro-glio-vascular unit work together to synchronize their energetic and electrical processes [120,121]. This section discusses how cognitive operations are supported by the coordinated operation of these different units and how they are able to use their molecular dynamics, redox-dependent phase relationships and frequency-dependent coupling between electron transport and vascular supply to encode electrical and energetic processes in a way that is temporally aligned.

### 4.1. Metabolic Microcircuits and the Predictive Economy of Energy

Unlike purely reactive networks, cortical networks operate in a predictive manner. It appears that metabolic supply is increased based upon recent network activity and inferred demand prior to a ~200–300 ms delay between synaptic bursts and perfusion increases. Astrocytes are central to this predictive process; synapse proximal astrocytic mitochondria measure ADP/ATP fluctuations and generate NADH oscillations that regulate glycolytic throughput via feedback mechanisms such as phosphofructokinase control [27]. Lactate is released in a frequency-structured manner, where rapid micro-bursts of lactate are associated with high-frequency (i.e., gamma band) activity and longer infra-slow components of lactate release sustain baseline states, providing distinct metabolic profiles for fast processing and persistent network activity [122].

Electrical activity is inherently coupled to metabolic load. Action potentials require temporary ATP demand and activation of mitochondria, resulting in localized heat and redox changes. During transmitter uptake, astrocytic Na^+^ influx drives Na^+^/HCO_3_^−^ cotransport and pH microdomain changes that increase glycolysis via pH-sensitive enzymes such as phosphofructokinase [123]. The phase lag between electrical depolarization and substrate mobilization (typically ~120–250 ms) corresponds to the gamma-band correlation window, indicating that bioenergetic kinetics are encoded in the temporal coding of cognition [124]. Astrocytic endfeet convert ionic fields into vascular tone; K^+^ efflux through large conductance channels hyperpolarize mural/endothelial membranes, and when combined with Piezo1-dependent shear-Ca^2+^ entry, modulate dilation kinetics and timing [125]. Stable hemodynamic phase lag constants have been confirmed by functional ultrasound across cortical territories, consistent with a conserved time scale for neuro-metabolic coupling [126,127].

### 4.2. Redox Phase Alignment and Quantum Efficiency of Electron Flow

Mitochondrial populations in neurons and glia display organized oscillations in NADH and flavoprotein autofluorescence indicative of redox cycling rather than random fluctuations. These rhythms arise from the coordinated unit activity of the respiratory chain and can demonstrate millisecond-scale phase coherence within mitochondrial populations [128]. Collective redox cycling increases electron transfer efficiency and ATP yield per O_2_ molecule by decreasing energetic losses due to redox transitions. Furthermore, mitochondrial phase dynamics are related to vascular oxygen delivery [129]. Elevated oxygen tensions can temporarily disrupt mitochondrial phase relationships; perivascular astrocytes may restore redox order through local NO/ROS-dependent tuning, thus maintaining oxidative efficiency and synchronizing oxygen delivery to ongoing computational demands [130]. In this framework, astrocytes provide not only metabolic substrate handling but also redox stabilization control at the gliovascular boundary [131].

### 4.3. Metabolic Field Potentials and Frequency-Dependent Resonance

Slow electrophysiological potentials have been observed to continue even after synaptic blockade, which has led to interpretations of metabolic field potentials generated by collective ionic currents linked to glycolytic and mitochondrial cycling [132].

These energetic potentials correlate with vasomotor fluctuations, indicating that electrical and metabolic fields represent coupled outputs of a shared oscillator. Cross-frequency analysis indicates that infra-slow hemodynamic rhythms modulate higher frequency neuronal activity (e.g., gamma-band), consistent with principles of amplitude/phase modulation whereby slow metabolic oscillations influence the threshold for excitation of fast neural assemblies [133]. Using two-photon phosphorescence imaging, it has been demonstrated that perivascular astrocytic Ca^2+^ bursts coincide with transient oxygenation spikes and elevated mitochondrial membrane potential, indicating that efficient computation occurs when all three oscillators are phase locked. When these phases are decoupled (due to fatigue, inflammation, or endothelial stiffening), computation can occur but becomes energetically expensive, evident by increasing cost-per-spike and behavioral slowing [134,135].

### 4.4. Adaptive Homeodynamics and Predictive Stability

The homeostatic function of neurometabolic coupling is characterized by multi-layered regulatory control that adjusts gain and frequency to maintain phase coherence between computation and supply. Rapid Ca^2+^-lactate microloops align demand with initial dilation, intermediate redox/oxygen feedback maintains coupling over seconds, and slow transcriptional programming sets baseline operating points over hours [136,137]. AMPK in astrocytes senses shifts in the AMP/ATP ratio, regulating metabolism towards rapid glycolytic support and instructing endothelial SIRT1/3-dependent antioxidant/compliance regulation, while pericytic mechanotransduction phospho-programming (including focal-adhesion kinase signaling) regulates contraction thresholds to maintain stable flow tracking under varying demand [138,139]. Disruptions in coupling (hypoxic microzonation, glial energetic fatigue, endothelial stiffening) are introduced into the system primarily as phase noise, indicating that cognitive resilience is heavily dependent on the maintenance of timing control [140]. Table 2 provides a summary of the hierarchical organization of neurometabolic-electrophysiological coupling across scales.

### 4.5. Integrative Perspective

Ionic signaling, metabolism and circulation are not independent processes but are interdependent oscillatory layers of a single control manifold. As such, the gliovascular network supports cognition by coordinating molecular oscillations that predict and stabilize energy delivery during computation, rather than simply supplying based on demand. Closed-loop resonance between ionic motion, metabolic cycling and vasomotor rhythm transforms cortical activity into a coordinated energetic field where blood flow adjustments track the temporal structure of ongoing information processing [151,152].

Critical Discussion and Alternative Viewpoints

There is debate regarding whether predictive neurometabolic control is primarily driven by astrocytes (via lactate/redox-gated signaling) or is primarily a result of neuronal signaling and endothelial autoregulation. One viewpoint emphasizes that astrocytes are the primary predictive integrators of demand and instruct vascular supply through glycolytic-lactate and redox oscillations [153]. An alternative viewpoint proposes that rapid energy flow alignment is primarily determined by direct neuronal metabolic demand signals and endothelial/pericytic intrinsic vasomotor/autoregulatory programs, with astrocytes making greater contributions to lower frequency gain control and substrate redistribution. Current evidence supports contributions from both viewpoints with the degree of contribution dependent on brain region, vessel type and behavior state [154]. The framework provided in this review is designed to combine these pathways into a timing-based model that provides testable predictions about the context-dependent prevalence of each pathway.

## 5. Interoceptive Regulation and the Hierarchy of Brain–Body Synchrony

Rather than being stabilized independently, cortical computations are continuously stabilized through their interaction with the physiological systems of the viscera. The neural circuits, glial networks, and vascular oscillators involved in such stabilization are synchronized with the cardiac, respiratory, enteric, endocrine, and immune rhythms, creating a bidirectional neurometabolic interface between cortical energetics and systemic homeostasis that represents a continuum of reciprocal resonance between brain and body oscillators at multiple timescales [155].

### 5.1. Visceral Oscillators and Cortical Phase Entrainment

Cardiac and respiratory rhythms provide a low-frequency spatial structure for cortical excitability. Studies using electrocorticography and functional MRI have demonstrated that heartbeat-evoked and respiration-phase signals modulate neuronal responsiveness in the amygdala, insula, cingulate, and somatosensory cortices through baroreceptive, vagal, and glossopharyngeal interoceptive pathways as opposed to mechanical artifacts [156]. The afferent volley associated with each beat can reset the thalamo-cortical phase, thereby aligning excitability windows with the pulsatile delivery of oxygen and glucose. The additional entraining axis provided by respiration further regulates cortical function by increasing noradrenergic tone through olfactory bulb-locus coeruleus coupling during inhalation and facilitating attention/memory retrieval, whereas exhalation promotes a shift towards parasympathetic dominance and internal processing [157]. Even under conditions of isocapnic breathing, cortical oxygenation has been found to vary with respiratory phase indicating direct pulmonary-cortical entrainment [158]. Astrocytes in the medulla/pons have been shown to express CO_2_-sensitive connexin-dependent signaling that links systemic gas exchange to central excitability, representing an extension of coordinated cardiorespiratory-neural coupling (“triad”) [159]. Under normal physiological conditions these oscillators tend to synchronize at rational frequency ratios (e.g., 1:4, 1:8) providing a global energetic scaffold; however, during stress/disease these ratios often deviate from those expected under normal conditions, suggesting that synchrony metrics may represent early markers of resilience/vulnerability [160,161].

### 5.2. The Vagal Axis and Immunometabolic Signaling

The vagus nerve provides the primary conduit by which glia communicate with peripheral organs. Afferent fibers of the vagus transmit cytokine levels, metabolites, and visceral stretch to the nucleus tractus solitarius, where they report on the integrated metabolic/immune state; efferent fibers of the vagus modulate spleen, liver, and gut activity and generally reduce inflammation and regulate substrate flux [162]. Thus, the closed-loop nature of this architecture allows for the link between inflammatory tone and cerebral energetics while also allowing for the output of cortical/autonomic activity to influence systemic immune activity [163]. Astrocytes of the dorsal vagal complex express pattern recognition receptors responsive to circulating cytokines, and the activation of glia results in the release of ATP and prostaglandins that can modify synaptic gain in autonomic nuclei and quickly influence heart rate and microvascular perfusion [164]. Conversely, cortical vagal drive influences microglia through alpha7 nicotinic cholinergic signaling, suppressing pro-inflammatory transcription and promoting mitochondrial biogenesis to maintain coupled energetic-immune stability [165,166]. Stimulation of the vagus has also resulted in measurable changes in circulating metabolomics (e.g., intermediate metabolites of the kynurenine pathway, sphingolipid metabolites, short-chain fatty acids) that can influence the metabolism of astrocytes/endotherlia and thereby alter the NVC gain states [167]. Metabolic/immune signals thus participate in repeating brain–viscera feedback loops that adapt cortical energetics to states [168].

### 5.3. Gut–Brain Energetics and Microbial Signaling

Bioactive metabolites produced by gut microbiota can reach the gut–liver–brain axis. Bioactive metabolites produced by gut microbiota, such as short-chain fatty acids (SCFAs), tryptophan metabolites, and secondary bile acids, can modulate microglial and astrocytic physiology. SCFAs, butyrate and propionate, can increase acetyl-CoA availability and influence mitochondrial efficiency and gene regulation through histone deacetylase-linked pathways [169,170]. Indole derivatives can act on endothelial aryl hydrocarbon receptors to modulate blood–brain barrier permeability and central metabolic programs, potentially linking nutrient state to circadian behavior [171].

The enteric nervous system serves as an independent oscillator coupled to brainstem circuits via vagal and sympathetic pathways. Interstitial cells of Cajal produce rhythmic potentials that organize intestinal motility and afferent signaling [172]. Enteric glia form connexin-linked networks and release ATP, NO, and neuropeptides that regulate inflammatory tone, similar to that of central gliovascular functions [173]. Activation of enteric glia correlates with plasma cytokine oscillations and can modulate insular astrocytic Ca^2+^ wave frequency via vagal feedback, suggesting an extended regulatory circuit connecting intestinal metabolism, immune tone and cortical energetics [174]. Endocrine and circadian rhythms provide slower temporal structures to these interactions. Cortisol and catecholamine oscillations regulate glial glycogen turnover, mitochondrial biogenesis, and vascular responsiveness; early waking cortisol mobilizes glucose in the liver and activates glycogenolysis in astrocytes through glucocorticoid receptors, priming substrate supply before behavioral activation [175]. Melatonin reduces mitochondrial redox potential and strengthens antioxidant buffering during sleep-linked repair. Endothelial clock genes (BMAL1, PER2) regulate time-of-day sensitivity to NO and pericyte contractility, while astrocytic metabolic enzymes (e.g., LDH, pyruvate kinase) display peak expression at different circadian times, regulating cortical energetic efficiency over rest–activity cycles [176,177]. Disruption of the circadian rhythm (shift work, insomnia, inflammation) disperses central and peripheral phases, reducing metabolic efficiency and leading to fatigue prior to the onset of structural injury [178].

### 5.4. Multi-Organ Coupling as an Energetic Network

In aggregate, the brain represents a node in a larger oscillatory network consisting of the heart, lungs, liver, intestine, and immune system. Rhythms in the heart define rapid hemodynamic windows; respiration modulates gas/ion coupling; hepatic glucose flux defines slow energetic baselines; and metabolites derived from microbes modulate long-term oxidative efficiency [179,180]. The interactions described above span > 6 orders of magnitude in time (ionic events in ms → circadian endocrine cycles), yet can remain phase coherent through predictive synchronization. Brainstem nuclei combine interoceptive signals to regulate cortical readiness prior to changes in systemic variables, while cortical networks impose descending rhythmic templates that determine the timing of visceral oscillators [49,181]. When the coherence between the brain and body fails (i.e., chronic inflammation, stress, metabolic disease), timing miscalibration between the brain and body (“temporal dysautonomia”) may occur prior to the manifestation of overt pathology [182]. Figure 2 illustrates this hierarchical-reciprocal organization of brain–body synchrony.

Critical Discussion and Alternative Viewpoints

A current debate exists regarding whether visceral rhythms serve to gate cortical computation or merely modulate it as a background signal interpreted by the cortex. On one hand, there is support for significant interoceptive phase setting in which cardiac/respiratory cycles reset thalamo-cortical excitability and limit cognitive timing to maximize supply–demand alignment [180]. On the other hand, there are also proponents who argue that many heart/respiration locked cortical signals represent state-dependent arousal, sensory reafference, or vascular confounds rather than causal phase control; in this perspective, cortical networks primarily dictate visceral patterns with interoception serving as a contextual modulator [181]. As evidence currently suggests bidirectional relationships with context-dependent weighting, we regard brain–body synchrony as a distributed timing system in which both visceral oscillators and cortical/gliovascular dynamics are mutually entrained, resulting in testable predictions of when interoceptive phase control will be dominant (e.g., stress, sleep, inflammation).

## 6. Molecular Plasticity and the Energetic Memory of the Brain

More than merely the reorganization of connections between neurons, the flexibility of the brain depends upon the continuous recoding of its energetic architecture. For each stimulus of neural activity, the energetic architecture is recoded in a manner that produces changes in the topography of mitochondria, the state of chromatin, and the geometry of the vasculature, producing a means of encoding energetic plasticity [183]. Thus, energy metabolism is no longer simply a passive fuel supply, but rather an actively optimizing, experience-responsive network. To concisely illustrate how the brain develops a stable energetic field over time, we will now describe three levels of constitution of this phenomenon, namely, epigenetic coding, organellar dynamics and redox reprogramming [184].

### 6.1. Epigenetic Energy Encoding: From Transient Flux to Enduring Regulation

Glial and endothelial cells undergo constant alteration in their transcriptional landscapes due to the metabolic activity they sustain. These alterations occur via the inscription of changes using NAD^+^ and AcCoA-dependent enzymic systems (for example, sirtuins, histone acetyltransferases, methane transferases), leading to permanent and enduring marks on the chromatin structure [185]. The transient bursts of oxidative demand caused by the repeated excitement of states produce SIRT1/3-mediated de-acetylation of mitochondrial transcription factors and increases in both the networks of organelles and oxidative demands. The repeated states of excitement, thereby, solidify these responses through remodeling of H3K27me3 and H3K9ac within the nuclei of the astrocytes, establishing a metabolic imprint at the level of chromatin structure for the preceding patterns of excitatory events [186]. The endothelial cells exhibit frequency-specific adaptation of transcriptional components, wherein oscillatory shear stress is expressed in consonance with periodic expression of KLF2 and eNOS and therefore expressed in consonance with the rhythm of blood flow. The adaptations observed are manifestations of longer-term responses to the imprints produced by the immediate denotations that result in an adaptation of vascular responsiveness to patterns of cortical oscillatory firing [187]. Such an epigenetic plasticity links the energetic history of the organ to further enhancements of computational processes. Astrocytes subjected to high-frequency synaptic activity, in a chronic situation, develop an induced up-regulation of the activities of several glycolytic enzymes and the capacities for transport of lactic acid, thereby lowering the threshold for an instantaneous and rapid production of energy. Conversely, underutilization, or a chronic situation, develops in a repressive chromatic complex of a reduced transcriptional capacity of oxidizing enzymes, thereby establishing an energetics phenotype of low efficiency [188]. These adaptations are mutually and reciprocally beneficial, and, thus, the energetic infrastructure of the brain can learn statistical characteristics of its own activity and predict the future requirements of the system, providing a molecular basis for the phenomenon of energetic anticipation and a natural accompaniment of synaptic prediction in neural circuits. There is co-repair of the vascular structures [189,190]. The continuous astrocytic VEGF and Wnt signaling remodels capillary density relative to the temporal statistics of the activation of the local tissue, whereas pericytes assume alternating phenotypes of contractile and permissive based on the flow dynamics. Therefore, the result of this process is a self-organizing energetic infrastructure, where the geometricization and metabolization of the vascular system tends toward a minimal phenomenon of self-entropy neurodynamic production relative to the correlated cognitive requirements of the local activated situation [191].

This structural metabolism resides in the area of feedback on the relationship of physical constraints through biochemical mutability. Mechanical stress sensed by pericytic Piezo1 channels alters the fission–fusion cycles of the mitochondria, thereby linking the mechanical interfaces of the tissue to productive sub-cellular energetic approaches. Furthermore, endothelial cytoskeletal alignment provides further support to directional delivery of oxygen, creating an anisotropic perfusion field that is adapted to prior use. Each of these aspects constitutes a part of a distributed learning algorithm that motivates the minimization of the energetic overheads of computation [192].

### 6.2. Redox Recalibration and the Adaptive Stability of Energy Flow

The cognitive condition of functioning consciousness requires an exactitude of control of the redox potential AP = aE—the differential of supply of electrons and demand for them—which is representative of metabolic homeostasis. This homeostasis is continuously self-optimized by the brain through the dynamic recalibration of its oxidative stress sensors and storage of antioxidants. The mitochondrial production of ROSs serves as a signal and perturbation. Thus, transient oxidation of all cysteine residues on ion channels affects excitability, whereas slight redox perturbations serve as an indicator of transcriptional upregulation of NRF2-mediated glutathione, thioredoxin, and peroxiredoxin systems [193]. Repeated controlled exposure to oxidative transients modifies the conditional response of these biochemical systems, resulting in redox conditioning—a biochemical analogue of tolerance, thereby increasing the operational range of the cortical metabolism. This adaptive stabilization extends beyond the cell. Glial nets communicate redox states through the diffusion of reduced glutathione via gap junctions and extracellular vesicles containing oxidoreductive enzymes [194]. Bursting of ROS in endothelial cells occurs in time with calcium waves in astrocytes, thereby ensuring that dilation of blood vessels coincides with release of antioxidants [195]. The entire gliovascular sheet constitutes a redox lattice, i.e., a field of coupled oscillators that dissipate oxidative stress through phase-locked feedback. When one node senses an overload of output of ROS, other nearby nodes decrease their temporary output of mitochondria, redistribute the energetic burden, and inhibit the qualitative inequality in loading [196]. The continued reorganization of this lattice represents the basis for the system’s ability to resist the ravages of aging, hypoxia, and trauma. Chronic training results in a baseline increase in the output of mitochondrial superoxide dismutase and uncoupling proteins generated by intermittent hypoxia, exercise, or cognitive challenges, thereby inhibiting the entropy costs associated with the generation of ATP [197]. A redox lattice that shifts toward a high-efficiency redox lattice gives the biochemical signature of an experienced brain, capable of learning by repeated exposure to fluctuations, and thereby working more efficiently with its machine of oxidation. Cognitive endurance does not derive from increased stores of energy but rather from the optimization of fluctuation, from a moderate instability that prevents catastrophic loss of synchrony [198].

Integrative Perspective

Mechanical translation of energy regulation into memory is referred to as molecular plasticity. Scientific conservation of the past of metabolism is achieved through epigenetic information [199]. Transmission of this information to forms of spatiality is provided by the organelle nets, whereas preservation of this information in the form of adaptive stability is provided by the redox reordering of reaction. These represent a higher hierarchical constitutional level of energetic learning, where each fluctuation of neural need leaves a trace in the substances of the system, and each trace alters the efficiency of the future. The brain viewed from this didactic perspective is not a static organism that consumes oxygen and glucose but a thermodynamically living energetic machine that writes new parameters for its function to preserve coherence [200]. The brain scientifically re-interprets neuroplasticity in terms of the energetic self-optimization of all the efficiency of the working parameters of its own function. Events occur kinetically in the synapses, but the conditions, the synthetic states of affairs, the mechanical stresses, and the rhythmicities of the metabolic systems in which the same events occurred are learned, stored, and preserved in the mitochondria, glia, and vessels. The faculty for memory enables the physical sustainability of cognition over the generations during which it is utilized. Perhaps within this quietly and continually revised code is to be located the most important, the most fundamental kind of learning, the faculty of the brain to learn how to power itself [201].

## 7. Network Thermodynamics and the Physics of Cognitive Stability

Thermodynamic principles govern how the brain maintains its own order despite continuous energy flow. An open, non-equilibrium biochemical reactor, the brain uses the molecular gradient of nutrients to generate information (as distributed in space). The energy dissipated in the process of generating information follows a statistical distribution, which the brain takes advantage of economically to maintain its own coherence as it organizes itself in the direction of least energy surprise. The subject of this paper will be the hidden architecture—the thermodynamics of thought—where entropy, forecasting, and biological time become a single, dynamic continuum [202].

### 7.1. The Brain as a Quantum–Thermodynamic Continuum

At the intersection of quantum and classical thermodynamics lies the metabolic base of the brain. Electrons passing through the respiratory complexes in the mitochondria demonstrate coherent times on the order of picoseconds. Such coherent times are sufficiently long to enable substantial metabolic efficiency in the organism. Experimental evidence indicates that the phase-coherent transfer of electrons through cytochrome arrays minimizes the amount of energy needed to induce stochastic reorganization of events (i.e., quantum metabolic coherence), thus maximizing the yield of ATP [203]. Coherence is fragile and efficient, and is lost when subjected to large oxidative loads but re-emerges during recovery sleep, suggesting an adaptive cycle of coherence/decoherence linked to circadian metabolic phases [204,205].

Mitochondrial ensembles also act at higher scales as amplifiers of stochastic resonance. Their membrane potentials oscillate in synchrony with cortical oscillations, thus modulating the kinetic rate of electron transfer. As a result, the brain exhibits resonance at multiple scales: quantum coherence of electron transfers is synchronized with collective oscillations of vascular and neural fields across approximately six orders of magnitude in frequency [206]. Energy is therefore transmitted with a temporal structure, i.e., energy quanta vary with oscillatory cycles of computations, ensuring that both molecular and cerebral functions have phase-compatible functions [207].

As an entropy motor, the brain converts micro-scale quantum uncertainty into macro-scale predictability. With each synaptic impulse, the brain navigates this scale-invariant process from single-electron displacement in respiratory chains to centimeter-scale vasomotor waves, all of which are regulated by the same principle: to reduce free energy by synchronizing their lengths of dissipation [208].

### 7.2. Thermodynamic Symmetry, Predictive Flow, and the Architecture of Entropy

Therefore, the economics of cognition is a function of symmetry, specifically a perfect correspondence between the creation of information and the release of heat. Neural systems seem to be able to maintain a nearly constant ratio between the entropy released as heat and the entropy eliminated via prediction. This ratio, the constant of information efficiency, approaches the Landauer limit within physiological limits, indicating that cortical computation is carried out at or near the lowest possible cost to eliminate information [209,210].

The gliovascular network is responsible for the maintenance of this symmetry because it is capable of actively redistributing gradients. Perivascular astrocytes are capable of detecting rapid changes in the density of entropy (quantal shifts in redox and osmotic pressures) and of triggering corrective fluid pulses to restore energy balance across the distribution of the gradient [211]. Small entropy waves generated by perivascular astrocytes permit the synchronization of local states of exchange without central coordination. Therefore, the brain continuously performs thermodynamic averaging on a volumetric basis to maintain an informational coherence through the localized variability of temperature or oxygen tension [212,213].

In terms of the system as a whole, this provides a physical interpretation of predictive coding systems: the hierarchical cortices minimize the production of entropy to match their internal models with the statistics of the peripheral environment. The metabolic correlate of this matching includes reduced variance in oxygen extraction fraction and redox potential. Efficient prediction reduces thermodynamic uncertainty: less surprise corresponds to less dissipation. Conversely, novel stimuli produce increased variance in energy produced by the form of entropy; however, controlled production of this type of entropy leads to learning. Therefore, learning is a type of thermodynamic excursion hyperactivity: a reversible excursion into disorder that produces future stability through reorganization of the excursion [214].

### 7.3. Energetic Phase Transitions and the Limits of Cognitive Resilience

It is the ability of the brain to maintain itself at the point of phase transition that allows for this stability. Cortical networks have empirically demonstrated convergence to subcritical and supercritical points of operation and the criticality necessary, such that even minor perturbations cause a reorganization of global states with minimal energy expenditure [215]. This metastable state is prepared for maximal information throughput in recovery methods, in thought as in turbulent flow of fluids, maintained at conditions just below the laminar break-up region [216].

This criticality is enabled by the action of the gliovascular system. The resistance of the microvasculature adjusts in real time to ensure that the constant flow of entropy through said networks remains unchanged, such that catastrophes of small-world synchronization or fracturing do not occur [217,218]. However, if the probability of energy contribution from metabolic activity exceeds the maximum permissible value, a phase slip occurs, which is a transient desynchronization of neural and redox oscillators observable as microstates of cognitive fatigue. Restoration of the original state of entropy requires corresponding vascular contraction and mitochondrial uncoupling, in order that the original gradients of entropy can be restored to the brain. Based on the arguments presented previously, these reversible micro-collapse states could be considered to be the physical basis for the temporary lapses of attention, or the brief periods of mental blankness of the mind of consciousness—the microcosmic or microscopic thermodynamic reset of entropy flows that reproduce in the state a sense of coherence on a global scale [219].

However, at the extreme limit of stressful inputs, the critical system may undergo catastrophic bifurcation: local collections of entropy create a topologically irreversible change in the flow of returning energy. Identification, traumatic injury or neurodegenerative disease causes the fragmentation of mitochondrial networks and the loss of phasic fidelity of capillary oscillators, and, with this, the cortex is shifted to subcritical states of high energetic rigidity or rigidity of density of molecules [103,220,221]. Recovery requires the introduction of controlled chaos through a controlled perturbation of instability, as exemplified by hypoxia, intermittent, neuromodulative stimulation and/or neuromodulatory treatments using neuroprosthetic devices of a suitable type, to bring about a return to the same organization of the line cool corridor. It is this principle that provides the foundation for the emerging field of thermodynamic rehabilitation—the controlled modulation of gradient flux of energy to facilitate a return, through self-control, cognitively speaking, of the cognitive self-organizational system [222].

From this perspective, it is suggested by the same phase transition theory that gliovascular disconnection (desynchronization) could also represent a first step towards the development of pathological conditions, in particular in clinical scales. Endothelial electric discontinuities due to ischemia or prolonged hypoperfusion, contraction failure of pericytes and loss of the polarity of astrocytic end foots could introduce “phase slip” between capillary responses and neuronal demands; this way, cognitive decline might occur even if global CBF has not yet reached the viability thresholds for large reductions in flow, due to timing errors causing high costs-per spike and destabilizing the network’s criticality [223]. Early pericyte dropout, remodeling of basement membrane/ECM, mislocalization of AQP4, inflammation, and decreased ability to modulate vascular tone are signs of increasing loss of oscillatory fidelity at the interface between glia and blood vessels in states such as Alzheimer’s disease and other neurodegenerative diseases; this leads to increased cortical dynamics of subcritical rigidity in advance of large amplitude perfusion deficiencies [224]. Similarly, small vessel disease is characterized by stiffening of microvasculature and dysfunction of endothelium; both lead to delayed or noisily transferred signal for molecular signals to diameter change, leading to large-scale network fatigue in excess of absolute flow decrease [225]. In addition, after traumatic brain injuries, spreading depolarizations, damage to glycocalyx and rapid inflammatory fragmentations of the adhesion between pericytes and endothelium will quickly disrupt phase alignments; thus, explaining why cognitive symptoms are usually larger than the size of structural lesions, and recovery may be dependent upon regaining synchronization rather than simply regaining baseline flow [226]. Overall, these contexts support a translationally applicable hypothesis consistent with our model, namely, that phase slips in gliovascular timing may serve as a sensitive biomarker for early cognitive vulnerability, and benefits may arise from therapies that resynchronize coupled glial–vascular–metabolic oscillators, regardless of whether modest changes in mean perfusion are observed [227].

Integrative Perspective

Viewed through the lens of thermodynamics, the brain is no longer simply to be regarded as a reactive biological device responding to external stimulation; rather, it appears as a self-calibrating entropy positioning engine: a clock face structure continually evolving to preserve the improbable structure of low-entropy conscious perception in bulk in a dissipative universe of quantum states [228]. Each oscillation, each synaptic potential, and each molecular reaction represent a negotiation between degradation and order, after the process of attractors or rejection of information flow rates. In this way, there is a quantum coherence of the mitochondrial structures, redistributions of vascular entropy, the application of predictive structures or symmetry in the neurologically hierarchically metamorphic structures of favor, etc., all for the purpose of converting the uncertainties of the inequalities of energy losses that occur into the geometry of such awareness or state of consciousness of associative and meaningful experience [229,230].

It is this fundamental assumption that, as is observed, unifies physics and neuroscience under a single law of self-preservation, whereby cognition can only continue through the production of entropy changes and variations in gravitational flux and its related phenomena. This implies that there are limits to intelligence, not of a computational type of problem as proposed by Turing and others, but of a thermodynamic type of functioning—what exists as thought proper can only exist in conditions when energy varies or is dependent upon its own basis of the possibility of restoring the critical state of coherence, so that the differentials of ennervation-metabolic and cognitive dissipation versus production of futures remain unchanged [231]. In this state, consciousness becomes the structure of the most highly ordered dissipative devices/products presently known—a continuous reorganization of energetic flows that enables itself to resist decay by learning from the patterns of its own ennervative sinks [232].

## 8. Conclusions: The Continuity of Energy and Mind

We have attempted to construct a cohesive conceptual framework based on an amalgamation of multiple disciplines—molecular biology, clinical physiology, and thermodynamics—in support of our thesis that the operational foundation of all cognitive processes lies in the degree to which a living unit in a living organism is able to sequence the available energy of its environment in time. This theme of integration and unification is consistent with the emphasis placed throughout the data collected, that the brain is not made up of individual parts but is a single, continuously changing field, encompassing metabolic, electrophysiological, and physiological properties that are different expressions of a single phenomenon: the continuous equilibrium between order and disorder in the system. From this perspective, what we refer to as perception, memory, or consciousness may be viewed as higher-order manifestations of the universal tendency for synchronization of the flux of energy.

Although we did not attempt to assert definitive conclusions, we have sought to collect and combine many different views of the mechanism of thought into a coherent conceptual framework. We have taken the view that the transitive flow of energy is both the substrate upon which neural computation occurs and the method through which stability, predictability, and adaptability occur. Electron transport within mitochondria and cortical oscillations appear to maintain cognition through a hierarchical feedback loop of glia, vasculature, and molecular programs. At the same time, we recognize that the current account of thought is partially speculative, and concepts that were discussed here—including redox phase coordination, timing-dependent neurovascular regulation, and self-organized flow dynamics—should be considered as suggestions for further experimental exploration, rather than end points of theory. It is expected that subsequent investigations will treat the brain as a living medium whose stability and ability to compute depend on the preservation of the continuity of the flow of energy and its temporal organization.

Several experimentally accessible approaches to investigating the proposed mechanisms of cognition follow from this framework. The first direction is to quantify timing as a physiological parameter. High-speed, two-photon imaging, fast functional ultrasound, optogenetic perturbations, and genetically encoded metabolic and redox reporters could be used to measure how the phase delay between the Ca^2+^ microdomains in astrocytes, pericyte and endothelial responses, and NADH/FAD oscillations change with brain state, development and disease. A second direction is to use closed-loop paradigms to determine if phase realignment between the gliovascular compartments can improve the energetic efficiency of the network computation without causing significant increases in blood flow. Third, spatial and single-cell multi-omics that correlate with dynamic physiology should elucidate how regional “signatures” of metabolism are encoded at the transcriptomic, proteomic, lipidomic and mechanotransductive levels and how these programs are modified during learning, aging or chronic inflammation. Fourth, the role of extracellular matrix mechanics, basement membrane composition, and glycocalyx integrity in modulating flow-signal transfer functions is poorly defined; multi-scale nanoscale profiling and in vivo mechanobiology may provide insight into when mechanical coupling is beneficial versus detrimental. Fifth, theoretical models should formalize the gliovascular interface as a multi-scale control system with measurable phase variables, stability margins and attractor regimes that can be compared against time-resolved physiological data.

Additionally, these directions indicate potential applications. For example, in basic neuroscience, gliovascular phase metrics may become an essential component to interpretating hemodynamic signals in fMRI, functional ultrasound and optical imaging by considering vascular rhythms as informative components of the control signal rather than noise. In clinical application, the identification of phase-slippage in the timing of the interaction between neurons and glial cells may serve as a sensitive indicator of cognitive vulnerability in conditions associated with small vessel disease, traumatic brain injury, migraine, sleep deprivation and neuroinflammation, potentially before structural damage has occurred. Furthermore, therapeutic restoration of timing coherence would provide motivation for treatments including neuromodulation (e.g., vagal, trigeminal, or deep-brain stimulation), respiratory or cardiac entrainment, targeted vasoactive or metabolic agents, and mechanobiological approaches that protect the endothelial glycocalyx and pericyte attachment, and could be measured through their ability to resynchronize the oscillatory behavior of the network as well as traditional measures of flow. In addition to medical applications, principles of predictive energetic control based on timing may be applied to the design of bio-inspired computing architectures where stability and adaptability are achieved through the phase alignment of resources as opposed to simply increasing the amount of energy supplied.

Thus, the task is not to increase the number of metaphors but to acknowledge that physics already has a vocabulary for regenerative self-stabilizing systems and that the brain exhibits these laws in a highly complex manner. We conducted this synthesis with caution, recognizing that the phenomena of molecular plasticity, energetic learning, and thermodynamic conservation need to be treated precisely and continue to be subject to experimental validation. The objective of this review is to establish a conceptual framework for relating empirical findings rather than stating final conclusions and to propose that there is an intrinsic language of energy and time that underlies the extensive vocabulary of neuroscience. If these ideas are accepted, they will be through empirical demonstration. Ultimately, it may be clear that, in order to understand cognition, we need to consider not only the signals that are transmitted but the temporal order of the flow of energy that sustains those signals. Therefore, this review concludes not with certainty but with hope: that the study of the energy topology of the brain will assist in integrating molecular precision with conceptual durability, and contribute to a common understanding of how, despite the constant draw of entropy, matter can maintain coherent thought.

## Figures and Tables

**Figure 1 ijms-27-00069-f001:**
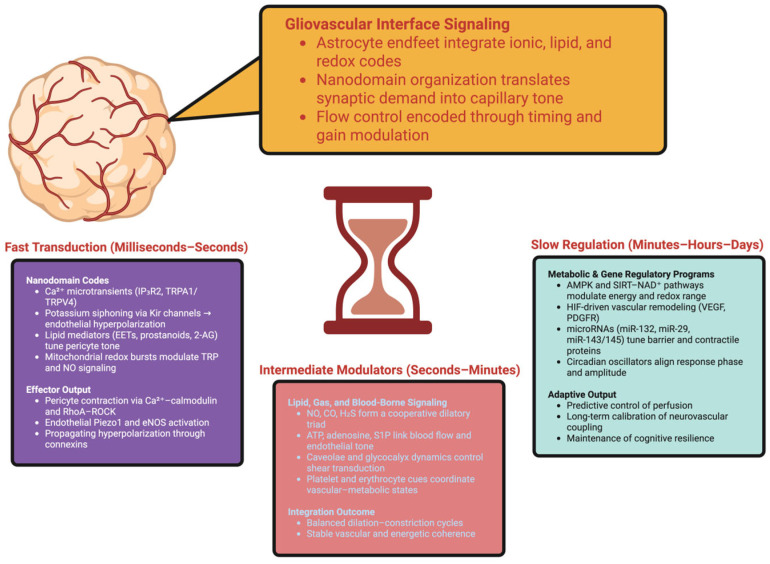
Molecular signaling at the gliovascular interface across temporal scales. This figure illustrates how astrocytic endfeet translate synaptic demand into vascular response by coordinating ionic, lipid and redox signaling. Rapid transduction mechanisms (milliseconds–seconds) involve Ca^2+^ microdomains, lipid mediators and electrical coupling, initiating rapid adjustments in pericytes and endothelial cells. Intermediate modulators (seconds–minutes), including nitric oxide, carbon monoxide, hydrogen sulfide and circulating mediators, continue and refine vascular tone through endothelial–mural integration. Slow regulatory programs (minutes–hours–days) governed by metabolic sensors, transcriptional networks, and circadian timing recalculate the interface for long-term coherence and cognitive resilience. Collectively, these layers comprise a hierarchical control system in which rapid nanodomain events trigger intermediate coordination and slow adaptive recalibration, maintaining temporal harmony between neural activity and blood flow.

**Figure 2 ijms-27-00069-f002:**
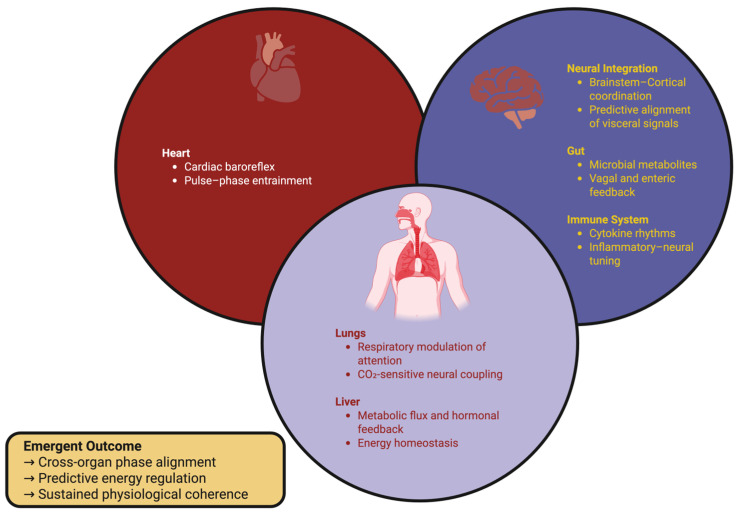
Hierarchical synchrony between brain and peripheral oscillators. The diagram illustrates how interoceptive regulation emerges from the coordinated activity of multiple organ systems. Cardiac and respiratory oscillations provide rhythmic scaffolds for cortical excitability and metabolic timing, while hepatic, immune, and gut-derived signals convey slower modulatory influences on cerebral energetics and inflammatory tone. Neural integration within the brainstem and cortical hubs aligns these visceral rhythms through predictive feedback, forming a bidirectional communication network that maintains energetic balance. The resulting outcome is cross-organ phase alignment, predictive energy regulation, and sustained physiological coherence.

**Table 1 ijms-27-00069-t001:** This table seeks to outline the multi-layered anatomy and signaling logic of the gliovascular unit, integrating structural, electrical, and metabolic data from cellular to network scales. It intends to condense key quantitative findings that define how astrocytes, endothelium, pericytes, and the extracellular matrix operate as a coherent signaling ecosystem.

Level/Component	Core Structural Features	Signaling and Functional Mechanisms	Quantitative/Experimental Data	References
Astrocytic Endfeet	Continuous perivascular sheath forming calcium and lipid microdomains; dense AQP4, Kir4.1, Cx43 nanoclusters	K^+^ siphoning, Ca^2+^ waves, gliotransmitter release (ATP, D-serine, glutamate); endfoot oscillations drive perivascular tone	[Ca^2+^]_1_ oscillations: 0.1–0.3 Hz; endfoot coverage ≈ 95% capillaries; AQP4 cluster radius ~50–80 nm	[65]
Endothelial Cells	Polarized membranes: luminal mechanosensors (PECAM-1, Piezo1); abluminal receptors (P2Y, CB1, prostaglandin)	Shear-stress Ca^2+^ influx, NO/cGMP signaling, bidirectional electrical propagation via connexins	Shear sensitivity threshold ≈ 1–2 dyn/cm^2^; conduction velocity 1–5 mm/s along endothelium	[66]
Pericytes	Contractile α-SMA^+^ cells with dendritic processes; tiled, phase-coupled architecture	Rho-kinase and Ca^2+^-dependent contraction; TRPV4 mechanotransduction; pannexin-mediated coupling	Contraction freq 0.1–0.2 Hz; Δdiameter ≈ 15–25%; mitochondrial density ↑ near soma	[67]
Extracellular Matrix (ECM)	Laminin- and agrin-rich basal lamina with dynamic sulfation gradients	Modulates ionic buffering, solute diffusion (D ≈ 10^−6^–10^−7^ cm^2^/s), and mechanotransduction; transient softening via MMP oscillations	Elastic modulus 0.2–1.0 kPa; MMP activity cycles ≈ 0.05–0.1 Hz	[68]
Cellular Polarity	Astrocyte: synapse → endfoot; Endothelial: lumen → abluminal; Pericyte: soma → distal processes	Intracellular Ca^2+^, NADH, and mitochondrial waves propagate directionally; coordinate feedforward-feedback vascular control	Ca^2+^ propagation speed ≈ 20–40 µm/s; redox oscillation phase delay ≈ 100–200 ms	[69]
Communication Modalities	Ionic (K^+^, HCO_3_^−^, IP_3_), chemical (ATP, NO, lactate), mechanical (matrix deformation), metabolic (NADH, FAD cycles)	Hierarchical coupling across temporal scales: ionic (~ms), chemical (~s), mechanical (~10^−2^–10^−1^ s), metabolic (~min)	Mechanical wave speed ≈ 10–30 µm/ms; NADH–FAD coherence R^2^ ≈ 0.8	[70]
Dynamic Remodeling	Activity-dependent vascular and glial morphogenesis; VEGF–Ang–Tie2 and ephrin-B2 signaling	Endfoot expansion, angiogenesis, pericyte remodeling under sustained neuronal activation	Capillary density ↑ 15–25% after 7 days of stimulation; AQP4 ↑ 40%; Cx43 ↑ 35%	[71]
Integrative Function	Coupled electrical–mechanical–metabolic network regulating local flow and systemic resistance	Synchronized oscillations align energy delivery with neural demand; global hemodynamic coherence as emergent property	Cortical flow oscillations 0.05–0.1 Hz; phase coherence Δφ < 10° between astrocyte–vessel units	[72]

**Table 2 ijms-27-00069-t002:** It aims to integrate structural, molecular, and dynamical dimensions of neurometabolic coupling across ten hierarchical levels—from nanoscopic redox oscillators to cortical field synchronization. It intends to highlight how ionic flux, substrate kinetics, and vascular motion operate as phase-locked components of a single oscillatory system. Quantitative parameters emphasize measurable frequencies, coherence indices, and bioenergetic efficiencies that define the thermodynamic precision of neural computation.

Scale/Level	Anatomical–Cellular Substrate	Core Biophysical Mechanism	Primary Molecular Mediators	Functional Outcome/Coupled Process	Quantitative/Temporal Parameters	References
Synaptic Microdomain	Perisynaptic astrocytic mitochondria, dendritic spines	Predictive energy release via ADP/ATP–NADH feedback loops	PFK, LDH, AMPK, NADH dehydrogenase	Phase-coupled lactate bursts support γ-band computation	NADH oscillation 0.1–0.3 Hz; lactate bursts 20–60 Hz; ATP recovery τ ≈ 200 ms	[141,142]
Astrocyte–Neuron Interface	Perisynaptic astrocytes, glycolytic clusters	Na^+^ influx drives astrocytic glycolysis through Na^+^/HCO_3_^−^ cotransport and pH-dependent enzyme kinetics	PFK, PK, MCT1/4, Na^+^/K^+^-ATPase	pH-linked energy gating aligns metabolic flux with synaptic demand	Bioenergetic phase delay 120–250 ms; ΔATP ≈ –10 µM/spike burst	[143]
Astrocyte–Vascular Interface	Endfeet with AQP4–Kir4.1–Cx43 clusters; endothelial mitochondria	Lactate/CO_2_ microbursts decoded via endothelial carbonic anhydrase and NO production	Carbonic anhydrase IV, eNOS, sGC, PKG	Rapid vasodilation and oxygen matching to neural frequency bands	Flow latency 150–250 ms; dilation amplitude + 15–20%	[144]
Pericyte–Endothelial Network	TRPV4^+^ pericytes and gap-junction–coupled endothelial cells	Electrical and mechanical coupling via K^+^ hyperpolarization and Ca^2+^ influx	Kir2.1, TRPV4, pannexin-1, Piezo1	Upstream propagation of vasomotor oscillations, pressure equalization	Conduction velocity 1–4 mm/s; Δdiameter ≈ 20%; oscillation 0.1–0.2 Hz	[145]
Mitochondrial Redox Network	Astrocytic and neuronal mitochondria across cortical layers	Coherent NADH/FAD redox oscillations; quantum tunneling synchronization of cytochromes	Complex I–IV, cytochrome c, CoQ10	Maximized ATP/O_2_ ratio via phase-coherent conduction	Redox phase coherence > 0.9; ΔATP/O_2_ ↑ 15%; entropy ↓ 5 × 10^−2^ J/mol K	[146]
Electrometabolic Cross-Talk	Astrocytic syncytium and interstitial ionic fields	Parallel flow of ionic and substrate currents; field potentials from glycolytic charge displacement	Kir4.1, Na^+^/K^+^-ATPase, H^+^ transporters	Electric and metabolic coherence—ionic activity entrains substrate flow	Potential–metabolic coherence R^2^ ≈ 0.75; propagation speed 0.1–0.3 m/s	[5]
Metabolic Field Potentials (MFPs)	Large-scale cortical and perivascular networks	Glycolytic wave interference with neural oscillations	NAD^+^/NADH, lactate, K^+^, Ca^2+^	Cross-frequency coupling of 0.03 Hz metabolic waves with γ (40–80 Hz)	Cross-modulation index ≈ 0.4; coherence R^2^ ≈ 0.8	[147]
Redox–Oxygen Feedback Loop	Astrocytic NO–superoxide system; vascular endothelium	O_2_ tension changes modulate mitochondrial synchronization through NO–ROS balance	eNOS, SOD2, COX4, glutathione peroxidase	Restoration of mitochondrial phase coherence and flow efficiency	O_2_ oscillation ± 3–5 mmHg; redox recovery τ ≈ 2–4 s	[80,148]
Homeodynamic Regulation Layer	Astrocytic AMPK–SIRT1/3, pericytic FAK, endothelial FOXO3A	Adaptive tuning of redox capacity, vascular stiffness, and oxidative load	AMPK, SIRT1/3, FAK, FOXO3A	Phase stabilization across molecular to network scales; predictive energetic control	Hierarchy: Ca^2+^ (Hz) → redox (10^−1^ Hz) → transcription (10^−3^ Hz); coherence loss ≥ 30 min pre-ischemia	[149]
System-Level Coupling	Integrated gliovascular–neuronal ensemble	Cross-scale resonance linking electrophysiology, metabolism, and hemodynamics	NO, lactate, BDNF, glutamate, astrocytic Ca^2+^	Unified oscillatory continuum sustaining predictive cognition	Flow oscillation 0.05–0.1 Hz; phase lag < 10° across networks	[150]

## Data Availability

The data presented in this study are available upon request from the corresponding author.

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
