# Peer review of "Neurovascular Signaling at the Gliovascular Interface: From Flow Regulation to Cognitive Energy Coupling"

_ijms, 2025, doi:10.3390/ijms27010069_

Round 1
Reviewer 1 Report
Comments and Suggestions for Authors
This manuscript is rather unusual. The broad idea of providing a theoretical framework for the gliovascular interface is interesting. However, the manuscript contains substantial conceptual, terminological, and factual inaccuracies. Many of the basic physiological concepts and mechanisms relating to NVC are described in rather unusual, not scientifically supported. There are many speculative interpretations without clear evidence. Here are some of the examples for the above comments:
- The statement that endfeet fully enclose every capillary segment is not correct.
- Is "quasi-homogeneous glial syncytium" meant to suggest that glial cells are similar to the functional syncytium like cardiac myocytes? Astrocytes do form gap junctions, but it is heterogeneous.
- Astrocytes do not function as a primary diffusion barrier. Astrocyte endfeet can modulate the BBB but they are not the BBB.
- The claim that blood flow is delivered in discrete quanta synchronized with neural oscillations is not supported.
- The reviewer is not sure how the interpret the following statements/phrases:
- chemical intelligencia
- factorial epoch
- humanitarianism
- metabolic synapses
- mechanical properties of the viscoelastic matrix charge-distribution
- magnitudes of mechanotransduction occurring under the mother membrane.
- and many more.....
Some of the terminology and phrases used are unusual in a scientific context.
Author Response
Dear Esteemed Academic Reviewer,
We are grateful for your rigorous and detailed assessment. Your comments were invaluable in helping us correct inaccuracies, clarify terminology, and sharpen the distinction between established physiology and theoretical synthesis. We revised the Introduction (Section 1) and the full anatomical/mechanistic core (Section 2) accordingly, and we also performed a broader sweep to remove similar issues elsewhere.
Comment:
The manuscript is unusual and contains conceptual, terminological, and factual inaccuracies regarding NVC, with speculative interpretations lacking clear support. Examples include: (i) the claim that endfeet fully enclose every capillary segment, (ii) the term “quasi-homogeneous glial syncytium,” (iii) framing astrocytes as a primary diffusion barrier/BBB, (iv) the idea of discretely quantized blood flow synchronized to neural oscillations, and (v) multiple unclear expressions (“chemical intelligencia,” “factorial epoch,” “humanitarianism,” “metabolic synapses,” ECM/mechanotransduction phrasing, and others).
Response:
We appreciate these corrections and agree with them. We addressed each point directly:
Endfoot coverage. We removed the absolutist statement and now describe astrocytic endfeet as providing extensive but heterogeneous, locally incomplete ensheathment of capillaries (Section 2.1).
“Quasi-homogeneous glial syncytium.” This term was deleted. Astrocytes are now presented as a gap-junction–coupled but regionally and phenotypically heterogeneous functional network.
Astrocytes vs. BBB/diffusion barrier. All wording implying endfeet are the BBB or a primary diffusion barrier was removed.
Quantized flow. We eliminated all claims of discretely quantized blood-flow delivery. CBF is now described as continuous and graded, though rhythmically structured and capable of frequency-dependent alignment with neural activity [17,18].
Unclear terminology. All listed nonstandard phrases were removed or replaced with conventional terms. We also removed additional similar expressions throughout these sections.
Evidence vs. speculation. We tightened narrative framing so established mechanisms are clearly separated from theoretical extensions, which are now explicitly presented as testable hypotheses.
We hope these revisions resolve your concerns and bring the manuscript in line with contemporary NVC/NVU physiology and terminology. We are sincerely thankful for your guidance, which substantially strengthened the work.
With respect and gratitude!!!
Reviewer 2 Report
Comments and Suggestions for Authors
This is an ambitious, highly integrative, and intellectually stimulating review that offers a novel conceptualization of the brain as an energetically synchronized, temporally phase-locked system. The emphasis on gliovascular coupling as a dynamic timing mechanism, rather than simply a supply-response system, is particularly original and valuable.
I only a few suggestions, 1) First, while the conceptual framework is highly sophisticated, portions of the paper rely heavily on theoretical interpretation rather than on direct experimental validation. it would be helpful to anchor these claims in more explicit empirical findings or cite specific recent studies supporting them. 2) the manuscript would benefit from clearer articulation of potential pathological implications of gliovascular desynchronization — for example in ischemia, Alzheimer’s disease, small-vessel disease, or traumatic injury. Since the authors suggest that cognitive vulnerability arises from timing failures rather than absolute metabolic deficit, extending the discussion into clinical contexts would significantly enhance the translational relevance of the review.
Overall, this is a strong and compelling work that deepens conceptual thinking in neurovascular biology and neural energetics.
Comments on the Quality of English LanguageThe manuscript is written in generally fluent and sophisticated English. The vocabulary is rich and expressive, and the authors demonstrate strong command of technical terminology. I only recommend simplifying certain passages to improve readability. Overall, the language is of good quality.
Author Response
Dear Esteemed Academic Reviewer,
We are grateful for your thoughtful, encouraging, and carefully reasoned evaluation of our manuscript. Your reading captured the intent of the work with remarkable precision, and we truly appreciate your generous recognition of the conceptual direction we aimed to develop. We also value your suggestions, which helped us strengthen the empirical grounding and the translational clarity of the review. Below we respond point-by-point and indicate the revisions made.
Comment 1: Portions of the framework rely heavily on theoretical interpretation; please anchor these claims in more explicit empirical findings or cite specific recent studies supporting them.
Response 1:
Thank you for highlighting this important balance. In response, we added a compact empirical-anchoring paragraph directly following our description of multi-layer timing regulation at the gliovascular interface. This new text explicitly links the timing-based framework to convergent experimental findings across scales, including: (i) high-speed in vivo imaging demonstrating sub-second coupling between astrocytic perivascular Ca²⁺ microdomains, mural-cell tone changes, red-blood-cell velocity, and capillary diameter; (ii) evidence for intrinsic vasomotor oscillations in pericytes and capillary segments that persist under synaptic blockade and can entrain neighboring microvessels via endothelial electrical conduction; and (iii) systems-level demonstrations of cross-frequency coupling between infra-slow hemodynamic waves and cortical electrophysiological rhythms.
Comment 2: Clearer articulation of pathological implications of gliovascular desynchronization (ischemia, Alzheimer’s disease, small-vessel disease, traumatic injury). Extending into clinical contexts would enhance translational relevance.
Response 2:
We are very thankful for this suggestion and agree that the clinical implications should be more visible. We therefore expanded Section 7.3 by adding a dense translational paragraph that interprets gliovascular phase-slippage as an early pathological mechanism across several contexts.
Once again, we appreciate your supportive assessment and the clarity of your recommendations!!!
Your comments strengthened the manuscript in ways that are both scientifically and clinically meaningful. We hope the revised version now reflects the balance you advised maintaining conceptual boldness while being more explicitly grounded in empirical evidence and translational context.
Reviewer 3 Report
Comments and Suggestions for Authors
The article constitutes a comprehensive interdisciplinary review that synthesizes data from molecular neurobiology, vascular physiology and bioenergetics to formulate a unified concept of cognitive processes as coordinated energetic states of the brain. The topic is highly relevant in the context of contemporary brain research, which is transitioning from purely electrophysiological models to integrative approaches that account for metabolic, vascular, and glial components. The authors successfully demonstrate how the gliovascular interface functions as a dynamic system ensuring precise temporal synchronization between neural activity and energy supply.
Recommendations for article authors:
-
Condense and structure the most complex sections (3–5), adding concise summaries at the end of each.
-
Incorporate a critical discussion of contentious issues, outlining alternative viewpoints.
-
Expand the 'Conclusion' section with specific suggestions for future research and potential applications.
-
Proofread the entire text for stylistic and editorial consistency, and standardize the citation format.
-
The abstract is overloaded with complex terminology and should be made more accessible to a broader audience.
-
The citation style is inconsistent in places (e.g., in-text references appear as [27],[28] without spaces).
This article represents a significant contribution to contemporary neuroscience, offering an integrative model that can serve as a foundation for new experimental and theoretical research. Despite some speculative elements and complexity of exposition, the work is distinguished by its depth, interdisciplinary approach, and conceptual boldness. Upon revision addressing the provided feedback, the manuscript can be recommended for publication in the journal.
Author Response
Dear Esteemed Academic Reviewer,
We are grateful for your careful, generous, and conceptually insightful evaluation of our manuscript. Your comments helped us see more clearly where the presentation could be tightened, clarified, and strengthened for a broader readership. We appreciate the time you invested in engaging with both the interdisciplinary scope and the mechanistic nuance of the work. Below we respond point-by-point and indicate the revisions implemented in the manuscript.
Comment 1: Condense and structure the most complex sections (3–5), adding concise summaries at the end of each.
Response 1:
Thank you for this precise recommendation. We fully agreed that Sections 3–5 had grown too dense in places and would benefit from stricter organization. Our goal was to preserve the scientific detail while improving flow, legibility, and conceptual hierarchy.
Comment 2: Incorporate a critical discussion of contentious issues, outlining alternative viewpoints.
Response 2:
We appreciate this request and recognize its importance for an integrative review. To address it, we added dedicated “Critical discussion and alternative viewpoints” subsections within Sections 3, 4, and 5. In these paragraphs we explicitly highlight current debates (e.g., relative primacy of astrocytic versus neuronal/endothelial initiation of rapid NVC; predictive neurometabolic control driven by glial integration versus intrinsic neuronal/endothelial autoregulation; causal gating versus contextual modulation in visceral–cortical entrainment). We also clarified that our proposed timing-based framework is intended to unify these pathways into a single testable control model, rather than to claim exclusivity for any one mechanism.
Comment 3: Expand the 'Conclusion' section with specific suggestions for future research and potential applications.
Response 3:
Thank you for encouraging a more forward-looking conclusion. We expanded Section 8 by adding a specific future-directions paragraph that outlines tractable experimental programs (fast multimodal imaging of phase delays; closed-loop entrainment to test causality; alignment of spatial/single-cell multi-omics with dynamic gliovascular physiology; mechanobiology of ECM/glycocalyx transfer functions; formal control-systems modeling with identifiable phase variables). We believe these additions make the conclusion more useful for guiding future work.
Comment 4: Proofread the entire text for stylistic and editorial consistency, and standardize the citation format.
Response 4:
We are grateful for this reminder.
Comment 5: The abstract is overloaded with complex terminology and should be made more accessible to a broader audience.
Response 5:
We appreciate this point and agree that the abstract should welcome a wider neuroscience readership. We rewrote the abstract to be more compact and accessible, replacing overly technical or metaphor-heavy phrasing with clearer language, while preserving the central claims, scope, and novelty. The revised abstract now foregrounds the main message early, uses fewer nested concepts per sentence, and maintains a more direct explanatory tone.
Comment 6: The citation style is inconsistent in places (e.g., in-text references appear as [27],[28] without spaces).
Response 6:
Thank you for noting this concrete formatting issue.
Once again, we are thankful for your constructive guidance and for the encouraging assessment of the manuscript’s contribution. Your recommendations significantly improved clarity, balance, and accessibility while preserving the interdisciplinary depth of the review. We hope the revised version now meets the standards you outlined, and we remain grateful for the opportunity to refine the work under your thoughtful critique.